# Center-of-Mass *iso-*Energetic Collision-Induced Decomposition in Tandem Triple Quadrupole Mass Spectrometry

**DOI:** 10.3390/molecules25092250

**Published:** 2020-05-10

**Authors:** Federico Maria Rubino

**Affiliations:** LaTMA Laboratory for Analytical Toxicology and Metabonomics, Department of Health Sciences, Università degli Studi di Milano at “Ospedale San Paolo” v. A. di Rudinì 8, I-20142 Milano, Italy; federico.rubino@unimi.it

**Keywords:** amino acids, equation, HPLC, MS/MS, NTS techniques (separation, ionization, and detection), nucleosides, open access software, target gas, triple quadrupole

## Abstract

Two scan modes of the triple quadrupole tandem mass spectrometer, namely Collision Induced Dissociation Precursor Ion scan and Neutral Loss scan, allow selectively pinpointing, in a complex mixture, compounds that feature specific chemical groups, which yield characteristic fragment ions or are lost as distinctive neutral fragments. This feature of the triple quadrupole tandem mass spectrometer allows the non-target screening of mixtures for classes of components. The effective (center-of-mass) energy to achieve specific fragmentation depends on the inter-quadrupole voltage (laboratory-frame collision energy) and on the masses of the precursor molecular ion and of the collision gas, through a non-linear relationship. Thus, in a class of homologous compounds, precursor ions activated at the same laboratory-frame collision energy face different center-of-mass collision energy, and therefore the same fragmentation channel operates with different degrees of efficiency. This article reports a linear equation to calculate the laboratory-frame collision energy necessary to operate Collision-Induced Dissociation at the same center-of-mass energy on closely related compounds with different molecular mass. A routine triple quadrupole tandem mass spectrometer can operate this novel feature (*iso-*energetic collision-induced dissociation scan; *i-*CID) to analyze mixtures of endogenous metabolites by Precursor Ion and Neutral Loss scans. The latter experiment also entails the hitherto unprecedented synchronized scanning of all three quadrupoles of the triple quadrupole tandem mass spectrometer. To exemplify the application of this technique, this article shows two proof-of-principle approaches to the determination of biological mixtures, one by Precursor Ion analysis on alpha amino acid derivatized with a popular chromophore, and the other on modified nucleosides with a Neutral Fragment Loss scan.

## 1. Introduction

The discovery of collision-induced dissociation of gas-phase ions and the introduction of tandem triple quadrupole mass spectrometers [1,2] has revolutionized the art of trace organic analysis in complex matrices in the last forty years, and has opened the way to contemporary “*omic*” measurement of the complex and dynamic composition of biological compartments [3].

In particular, the two scan modes that, although not unique to tandem triple quadrupole [4], at which this configuration of mass spectrometer performs at its best are those known as Precursor (formerly, “parent” [5]) Ion (PI) scan and Constant Neutral Loss (CNL) scan. Their analytical strength allows selectively pinpointing, in a complex mixture, only compounds that feature specific chemical groups. The compounds can be identified as giving rise, upon fragmentation of the molecular ion, to a characteristic fragment ion (precursor ion scan) or to the loss of a characteristic substructure of the molecule(s) as a neutral species (neutral loss scan). These scan modes (Scheme 1) allow:(a)identifying families of structurally similar compounds that occur in a complex mixture;(b)assigning molecular masses to each of the components and(c)indicating the molecular precursors that can be characterized by a subsequent “fragment ion” scan (the earliest scan mode of tandem mass spectrometers) experiment to insight their molecular connectivity.

This analytical strategy can be automated in Data-Dependent modes of operation of commercial triple quadrupoles. Industrial pharmacology and toxicology found a particular benefit in applying this technique systematically to identify the products that derive from biotransformation of xenobiotics, such as candidate pharmaceutical drugs, pesticides, industrial chemicals. 

Several classes of conjugated xenobiotic and endogenous metabolites feature distinctive fragmentation pathways in their mass spectra, and selective scan experiments were elaborated to selectively detect the compounds that belong to each class of conjugates, such as glucuronides, sulphates, glutathione thioethers, mercapturic acids [6,7]. Specific scan modes have been developed to search for (and to discover, in the case of unexpected components) several endogenous classes of soluble metabolites, such as urinary acyl-carnitines [8], derivatized fatty acids and alcohols [9], complex lipids with a choline head-group, such as phosphatidyl-cholines and sphingomyelins [10], ceramides [4]. The Precursor Ion and Neutral Loss operation modes of the triple quadrupole tandem mass spectrometer allow the non-target screening of mixtures for structural classes, rather than for individual molecules, thus allowing the detection of new or unexpected components.

At the core of the application of tandem triple quadrupole instruments to the selective identification of chemical groups is the occurrence, in the fragmentation pattern of the molecular precursor(s), of specific decomposition channels with the following characteristics:(a)generate a sufficiently specific charged or neutral fragment as a reporter of the considered chemical class;(b)the considered Precursor-to-Fragment generation process (“transition”) occurs in the different molecules with a sufficiently constant kinetics (reflected by a fairly constant value of collision energy for maximum production);(c)the transition is sufficiently insensitive to the presence of other chemical arrangements in the molecule (reflected by a fairly constant value of relative intensity).

Therefore, the search for specific molecules of a defined structural class is performed by PI or CNL scan, in the triple quadrupole MS.

To operate Collision Induced Dissociation, the q2 (RF-only quadrupole collision cell) is kept at some value of the potential difference (in the instrument, the voltage drop, or collision offset voltage) with respect to that of the “first” quadrupole, considered at “ground” potential [11]. A “reasonable” value for this parameter usually corresponds to the maximum value of the production efficiency curve of the selected fragment ion that is recorded in one or more “typical” compounds as a preliminary measurement of the method setup. “Nominal” (or “laboratory frame”) collision energy, CE_lab_, is the product of the voltage drop and the number of charges of the precursor ion.

The efficiency of fragmentation, however, is not determined by the “laboratory frame” collision energy, but rather by that experienced by the isolated precursor ion undergoing collision with the stationary target gas, the “Center-of-Mass” collision energy; E_CM_. By operating at a constant difference of the (q2-Q1) potential (the “laboratory frame” collision energy), precursor ions of different m/z experience non-linearly decreasing amounts of additional internal energy, as their m/z value increases. In addition, the increasing molecular size increases the amount of energy that is necessary to achieve fragmentation (Degree-of-Freedom, DoF effect), as has been demonstrated in a wide-purpose systematic measurement of the “characteristic energy” (E_50%_) for fragmentation of organic polymers [12]. 

Very recent commercial instruments can adjust collision energy for fragment ion analysis of precursors with different m/z (especially of peptides in proteomic analysis) with the use of patented “mass-related” parameters [13], such as the Normalized Collision Energy [14,15], and otherwise-named approaches used by academic scientists [14,15,16,17,18]. However, it is likely that the instruments’ manuals do not explicitly report the equations employed to perform this task (e.g., http://proteomicsnews.blogspot.com/2014/06/normalized-collision-energy-calculation.html). In addition, the term “Normalized Collision Energy” is not even defined in the IUPAC 2013 “Definitions of Terms Relating to Mass Spectrometry” [19].

This article describes how to derive an equation to calculate the laboratory frame, or “nominal”, collision energy necessary to analyze closely related compounds with different molecular mass and using different gas targets for collision-induced decomposition tandem mass spectrometry. It further demonstrates that this novel feature (*iso-*energetic collision-induced dissociation; *i-*CID) can be applied in a standard analytical triple quadrupole tandem mass spectrometer, either by continuous ramping of the (q2-Q1) potential, or with a “stepped” surrogate, if the instrument’s software does not allow for continuous ramping. To suggest the utility of this application, the article reports two proof-of-principle analyses of mixtures of endogenous metabolites, one using a Precursor Ion scan, the other a Constant Neutral Loss scans. The latter experiment also entails the hitherto unprecedented synchronized scan of all three quadrupoles (Q1, q2, Q3) of the triple quadrupole tandem mass spectrometer.

## 2. Results

### 2.1. Theory and Simulations

In Collision Induced Dissociation of ions, the relation of center-of-mass, or “effective” (E_cm_) to laboratory frame or “nominal” (E_lab_) collision energy depends on the mass of the resting target gas in the collision cell (in most instruments Nitrogen, and occasionally Argon), (m TAR) and on that of the impinging precursor ion (m PRE), according to Equation (1) [20].

(1)$$E_{\mathrm{cm}}=E_{\mathrm{lab}}\times \frac{\mathrm{mTAR}}{\left(\mathrm{mTAR}+\mathrm{mPRE}\right)}$$

This transformation from “*laboratory frame*” to “*center-of-mass*” collision energy is widely employed to plot “breakdown curves” and to compare those obtained for precursor ion with different *m*/*z*.

For a fixed collision gas employed (in all described examples m TAR is Nitrogen, MW 28, or Argon, MW 40), and for each specific value of “laboratory frame” collision energy, the “center-of-mass” collision energy decreases as the *m*/*z* value of the precursor ions increase. The variation is not linear, neither with the *m*/*z* of the precursor ion, nor with the value of the “*laboratory frame*” collision energy (Figure 1).

The simulation displayed in Figure 1 refers, as a general example, to singly charged precursors ions of molecules that contain a common portion (molecular mass 150, in the example, corresponding to fragment ***f*** in the example of Scheme 1) and a poly methylene variable part with one to twenty carbon atoms (this originates the different fragment ions **A**, **B** and **C**). Collision-induced dissociation occurs on a target of Nitrogen gas and is calculated for increasing values of “laboratory frame” collision energy. The ion precursor dissociates after collision according to the following pathways:
**Precursor(C_n_)^±^****→****Const^±^ + (C_n_)^0^****Precursor ion scan****Precursor(C_n_)^±^****→****Const^0^ + (C_n_)^±^****Neutral fragment loss scan**

In the example of Figure 1, precursor ion Precursor (C_1_)^±^ (*m*/*z* 164 = 150 + 14) impinges on the Nitrogen gas target at a laboratory frame collision voltage of 20 eV and experiences a center-of-mass collision energy of 2.92 eV. At the same laboratory frame collision voltage, precursor ion Precursor (C_20_)^±^ (*m*/*z* 430 = 150 + 20 × 14) experiences a much lower center-of-mass collision energy of 1.22 eV.

The trend of the curves displayed in Figure 1 makes it possible to understand some literature results that have been reported for the measurement of molecular series by tandem mass spectrometry at constant laboratory-frame collision energy.

One such example in the literature is offered by the measurement of ten saturated, straight chain homolog esters of acyl-carnitine that span from C2 to C18, obtained by FIA-ESI and precursor ion detection of the *m*/*z* 85 fragment (C_4_H_5_O_2_; protonated butadienoic acid; Scheme 2), at 20 eV nominal collision energy onto an Argon gas target [8].

When the authors infused an equimolar solution of the standard compounds, the resulting Precursor Ion spectrum (Figure 3 of ref. [8]) showed a marked and regular decrease in signal strength of the lower (C-2 to C-8) and of the higher (C-16 to C-18) homologs. Figure 2, left panel, is recalculated from the intensities in the spectrum of the cited article.

In the right panel of Figure 2, the relative yields of the Precursor Ions are plotted against the center-of-mass collision energies, the values of which are in turn calculated for the respective molecular ions as corresponding to the employed laboratory frame collision energy of 20 eV against an Ar gas target, with the use of Equation (1). The bell-shaped trend can be interpolated with a parabola, the apex of which, at approximately 1.76 eV, yields an approximate estimation of the maximum of a fragmentation efficiency curve for the employed transition M^+^ ≅ 85^+^. This rough calculation also implies, but does not warrant, that under FIA (i.e., at a constant composition of the solvent) the ESI ionization efficiency does not discriminate homologs with an acyl chain of a greatly different length that are simultaneously present in the droplet.

This example shows that it is useful to have a systematic method to perform analyses by collision-induced dissociation and Precursor Ion or Neutral Loss scan in the triple quadrupole over a range of *m*/*z,* which keeps at a constant value the collision energy in the center-of-mass frame. To achieve this, the laboratory frame collision voltage needs being systematically increased as the *m/z* value of the precursor ion (m PAR) increases. 

Standard algebraic passages allow re-formulating Equation (1) to yield the linear working Equation (2):
(2)$$E_{\mathrm{lab}}=E_{\mathrm{CM}}+E_{\mathrm{CM}}\times \frac{m\;\mathrm{PRE}}{m\;\mathrm{TAR}}$$


This linear equation makes it possible to “back”-calculate the necessary laboratory frame collision voltage at each value of *m*/*z* of the precursor ion corresponding to a specific constant value of center-of-mass collision energy, as shown in Figure 3 for the same example of Figure 1.

Equation (2) can find use in transferring the collision-induced decomposition conditions that have been optimized for one specific molecular species within a class to other homologues with a similar fragmentation pattern, and to instruments that employ a different collision gas (e.g., Nitrogen in lieu of Argon, as seen above).

### 2.2. Proof-of-Principle Applications

The feasibility of the *iso-*energetic scan as an analytical tool to characterize complex mixtures of metabolites was tested with some proof-of-principle examples to understand whether this strategy can be useful in organic bio-analysis of small-molecule biomarkers.

Two experimental examples of application are reported. The first employs a precursor ion scan (the reporter molecular fragment is an ion, identified by the *m*/*z* of Q3), the second a constant neutral loss scan (the reporter molecular fragment is a neutral species, identified as the fixed *m*/*z* difference between Q1 and Q3).

#### 2.2.1. Detection of Dabsyl-Amino Acids by *iso*-Energetic CID and Precursor Ion Scan

The first proof-of-principle experiment of the use of *iso-*energetic collision-induced dissociation for the analysis of small organic molecules by Precursor Ion scan is obtained by derivatizing alpha-amino acids with a very popular chromogenic tag, dimethylamino-azobenzene (dabsyl, DABS-), linked to the amino-group as the sulphonyl amide (Scheme 3).

Some amino-acid derivatives were prepared individually (list in Appendix A) and the fragment spectra of their deprotonated molecules were measured in the negative ion mode over a range of collision energy up to 70 eV_lab_. Figure 3a,c shows as examples the integrated spectra recorded for the derivatives of the lowest- and highest-mass natural amino acids, glycine and tryptophan, respectively. The most intense fragment in the spectra, at *m*/*z* 240, is generated from the appended chromophore, as indicated in the general structure of the examined derivatives, and due to its common occurrence and high intensity qualifies as reporter ion for analytical purposes.

The generation efficiency curves of *m*/*z* 240 are displayed in the corresponding right-hand panels Figure 3b,d, and show the position of the maxima calculated as described in the Materials and Methods Section 4.5. In the laboratory frame, the values of the maxima are 29.50 eV_lab_ (center-of-mass frame, 2.12 eV) for DABS-Gly, and 35.66 eV_lab_ (center-of-mass frame, 1.93 eV) for DABS-Trp. Results of the measurements in a few other DABS-AA are collected in Table 1.

Least-squares calculation of center-of-mass collision energy vs. precursor ion *m*/*z* affords a best-estimate value of 2.15 eV for the representative value of CE_max_ for the characteristic [M–H]^−^ to *m*/*z* 240 transition (Appendix A). The obtained value corresponds to a scan-line that starts at 29.9 eV_lab_ for *m*/*z* 361 of deprotonated DABS-Gly and ends at 39.8 eV_lab_ for *m*/*z* 490 of deprotonated DABS-Trp (see parameters used in the example of Figure 1).

One interesting observation stems from the comparison of the scatter around the best-fit scan line of actual maxima measured in the individual production efficiency curves of *m*/*z* 240 (Appendix A). While differences as large as 3.9 eV_lab_ are found (the outlying Trp being that with the largest difference), this value should be compared to the actual span of the maxima of the round-topped curves recorded for these compounds. As can be appreciated from the right-column curves of Figure 4, the width of the 95%–100%–95% round-topped curve maxima is of 5 to 9 eV_lab_. The uncertainty of appreciation for the apex (the intersection of the two least-squares ascending and descending lines) results in the order of fractions of eV_lab_ (0.2–0.4 eV_lab_), less than the employed potential step in the accurate ramp-CE experiment (0.5 eV_lab_). Therefore, the method employed to obtain the representative value of center-of-mass CE_max_ yields a reliable value for method setup.

To test whether this experiment is amenable, and useful, to real applications in bio-analysis, a DABS-derivatized urine was analyzed with an un-optimized fast gradient elution linked to a 1-s *m*/*z* 240 precursor ion scan from 350 to 500 and to a synchronized scan of the collision voltage (CElab, q2-Q1) from −28.4 to −39.6 eV_lab_. Main results are displayed in Figure 4a–d.

Under the un-optimized “shotgun” chromatographic conditions of this experiment, several DABS-derivatized alpha-amino acids of urine elute as partially separated chromatographic peaks between 3.5 and 5.5 min (Figure 4a) and the integrated spectrum (merged spectra) of the corresponding chromatographic time-frame contains the molecular signals of the main physiologically expected amino acids (Figure 4b).

This un-optimized, very fast chromatographic condition does not separate the isomers of Leucine, nor isobaric hydroxy-proline, at *m*/*z* 417. The mono-DABS derivatized histidine signal is observed (*m*/*z* 441), and the signal at *m*/*z* 455 may correspond to the unseparated, isomeric methyl-histidines. Methionine can be identified from the co-occurrence of the molecular species at m/z 435 (32S) and −437 (^34^S). By analogy, a weaker signal at *m*/*z* 421 can be attributed to the trace amino acid S-Methyl-cysteine (SMC) [21], based on the appearance of a corresponding signal at *m*/*z* 423 of the corresponding ^34^S-species. This observation, in particular, opens the way to a non-target screening of cysteine thioethers in adductomics applications. No signals are observed, nor were likely expected, for those that yield a doubly dabsylated derivative, Cysteine and Lysine. Only one chromatographic peak is observed in the XIC of *m*/*z* 432 (Glutamine), and a very weak signal at *m*/*z* 460 is observed for mono-DABS-Arginine. Precursor Ion spectra extracted corresponding to the XIC for the main identified amino acids feature the expected [M-H]^-^ molecular signal expected for the individual eluting species (Figure 4c,d).

Identification of the amino acids by Data-Dependent Fragment Ion analysis was not performed, since all DABS-derivatives yield the same main fragment ions, all of which deriving from the appended chromophore (see Figure 4a,c).

#### 2.2.2. Detection of Nucleosides by *Iso-*Energetic CID and Constant Neutral Fragment Loss Scan

The second proof-of-principle example employs nucleosides, measured in the positive ion mode, to demonstrate that iso-energetic CID is compatible with Constant Neutral fragment Loss Scan (CNL) experiments in the triple quadrupole. In this experiment, all three quadrupoles scan synchronously; in fact, the q2-Q1 voltage difference is “ramped” simultaneously with the scan time of both Q1 and Q3.

Protonated nucleosides (listed in Appendix A) undergo collision-induced fragmentation through a main pathway that involves loss of the N-linked sugar as neutral (loss of 116 Da from the 2-deoxy-ribosides and of 132 Da from the ribosides) and formation of the protonated nucleobase (Scheme 4). A very minor pathway generates the complementary positively charged sugar species (*m*/*z* 117 and 133, respectively, for 2-deoxy-ribose and ribose) but their intensity is lower by more than two orders of magnitude with respect to that of the protonated nucleobases, therefore making a Precursor Ion scan utterly unsuitable for analysis.

As in the example before, the collision energy corresponding to the maxima of the production efficiency curves of the protonated nucleobases was measured from the fragment ion spectra (measurements in Table 2 and results of calculation of the representative CE_max_ in Appendix A). The maxima are between approximately 1.5 and 2.0 eV_CM_, with no analytically relevant difference between the 2-deoxy-nucleosides (loss of 116 Da; median 1.60 eV) and the nucleosides (loss of 132 Da; median 1.67 eV).

From the perspective of fast method setup, it is appealing to observe that the values of CE_max_ measured from the fragment intensity curves of the Fragment Ion spectra and those measured from the “reverse” experiment (CNL, in the case of nucleotides) are sufficiently close for the analytical application. As an example, Appendix A shows the curves for Guanosine (*m*/*z* 284 > 152, Fragment Ion spectrum) and (CNL132, *m*/*z* 284), both recorded from a mixture to ensure that calculation results derive from measurements of the same (noisy) quality. Estimated CE_max_ is 17.86 and 17.89 eV_lab_, respectively. 

That such close results can be obtained is advantageous when a few standard compounds are available. In this case, one single “reverse” experiment, rather than several recordings of individual Fragment Ion spectra, may still yield sufficiently reliable data to calculate a starting representative eV_max_ value for method optimization with the spreadsheet module of Appendix B.

To demonstrate the selectivity of the two CNL scans, Figure 5 shows the source spectrum of a mixture that contains the four DNA nucleosides (dA, dC, dG, dT) and two RNA nucleosides (only C and G; A was deliberately omitted, since it is isobaric to dG). The two inserts display the results of consecutive alternate 1-s CNL-scans of 116 Da (selective for deoxy-ribonucleosides) and 132 Da (selective for ribonucleosides) at 17 eVlab (1.6 eVCM for precursor *m*/*z* 270).

All expected signals that correspond to the protonated molecules of the six compounds are observed in the respective NL spectra. Adenosine was deliberately omitted from this mixture to confirm that the NL116 scan is specific for deoxy-ribonucleosides, since the precursor at *m*/*z* 268 from isobaric A is absent from the NL132 spectrum, but that from dG appears in the NL116 scan. 

A more complex mixture that contains the four DNA nucleosides (dA, dC, dG, dT), three RNA nucleosides (A, C, G), and six 6N-substituted derivatives of adenine (listed in Appendix A; complete ESI source spectrum in Appendix A) was further employed to investigate the potentiality of the iso-energetic CID scan for applications in the field of nucleoside analysis. In this mixture, the ratio of the isobaric dG:A is approximately 1:4.

The possibility of screening the sample for modified nucleobases is exemplified by six 6N-substituted derivatives of adenine (8–13) [22,23] with masses between 350 and 400 Da, and thus 30% to 50% higher than that of the generating riboside, adenosine. Their fragment spectra (one example is reported in Appendix A) feature loss of the ribose, and the first-generation protonated adenine further loses the N6-adducted substituent to yield the *m*/*z* 136 AdeH^+^ fragment, and often one from the appended molecular unit.

The experiment reported in Figure 6a–d is a concept with prospective real applications, such as the identification of natural epigenetic modifications of DNA nucleobases, of xenobiotic DNA adducts, of regulatory t-RNA modified bases, and of synthetic nucleobase tools for research in molecular biology [24]. Its aim is to evaluate whether the *i-*CID approach is feasible in a routine triple quadrupole instrument without compromising mass resolution and at a scan speed compatible with coupling to liquid chromatography. To test whether the use of a CE ramp improves the accuracy in the determination of nucleoside mixtures, the peak intensities of the non-isobaric nucleosides obtained in the fixed-CE and in the ramp-CE Neutral Loss scans were compared to those measured in the ESI source spectra of the same mixture. The cycle of measurement follows the pattern and conditions reported in Appendix A. In the same sample, a consecutive NL116_rampCE experiment, selective for 2-deoxy-ribonucleotides was also performed, with essentially similar results (data not shown).

As is apparent from the ion profiles (insert in Appendix A), no deterioration of resolution occurs when the q2 voltage is ramped synchronously with the Q1 and Q3 scans to achieve the collision energy scan. The fastest conditions chosen in the employed routine triple quadrupole instrument (infusion mode, CNL scan with synchronized q2 ramping) were a scan speed of 2.5 seconds/1.000 Da (*m*/*z* 220–420 in 0.5 seconds) and a collision voltage ramp speed of 23.6 eV_lab_/s (14.6–26.4 eV_lab_ in 0.5 seconds). To hint at the analytical potential of this experiment in bio-analysis, the relative signal intensity of the considered ribo-nucleosides was compared between the different analytical conditions, as summarized in Appendix A. In particular, the relative signal abundances of the substituted adenines were closer to those measured in the source spectra when the CEscan mode was employed, rather than at any fixed CE value (Appendix A).

In the employed routine triple quadrupole instrument (infusion mode), scan speed of the Neutral Loss scan with synchronized q2 ramping can be as fast as 0.5 s in a *m*/*z* range of 200 Da, without any consistent signal instability or loss of resolution (insert in Appendix A), and allowing for two scan modes to alternate (*i-*NL116 and *i-*NL132).

## 3. Discussion

The work described in this article shows that continuous variation of the collision voltage in the cell of a routine triple quadrupole instrument during a fast scan of the mass filter(s) is technically feasible both in the Precursor Ion and in the Constant Neutral Loss modes. 

This unprecedented operation of the TSQ can be used without degrading sensitivity and mass resolution, with respect to holding the collision cell at any intermediate constant value. This modification to the more common fixed-voltage practice allows collision-induced dissociation of molecules of a common structure but with different molecular mass to occur with each precursor ion facing the same value of effective (center-of-mass) energy deposition, and thus with the same, or close, relative kinetics. Therefore, when analyzing homogeneous series of chemical species, such as homologous series of metabolites, the intensity of corresponding fragments in different compounds will be more similar. Thus, the detection efficiency of the different homologs within a series improves, by overcoming the variation in collision efficiency that occurs when compounds over a wide range of m/z values are analyzed at a “compromise” value of laboratory frame collision voltage.

The convenience of this approach over the use of a fixed, “compromise”, value of collision voltage is more apparent when the “variable” part of the structure contributes more to the increase of molecular mass than the “fixed” one. In this case, the relative increase of molecular mass of the target compound with that of the “variable” part is larger, and so is the associated interval of collision voltages between the smallest and the larger molecules of the class. The other factor is the value of center-of-mass collision energy for the considered, analytically useful transition: a higher Center-of-Mass collision energy (slope of the CE scan line) leads to a higher interval of collision voltages for the same mass range.

This article examined two molecular classes with a different type of fragmentation as proof-of-principle, among the many different examples of possible applications. One class, dabsyl-derivatized amino-acids, feature a common reporter fragment ion in the spectrum, and can be detected by employing a precursor ion scan. The derivatization approach is still widely popular in many bio-analytical applications [25], and several molecular classes of natural compounds, such as complex lipids, are similarly featured, with a common charge-carrying substructure and variable modifications that increase the molecular mass [4]. Another class of examined molecules, ribosylated nucleotides, features a common decomposition pathway that eliminates as a neutral the common carbohydrate sub-structure, and yields the variable nucleobase as an ionic fragment, so a Constant Neutral Loss scan is the most adequate tandem-MS experiment to highlight this class of molecules.

## 4. Materials and Methods 

### 4.1. Reagents and Standards

All solvents, reagents and standards were of analytical purity (Prodotti Gianni, Milano, Italy), and were available in the laboratory from previous or current analytical and spectroscopic studies. Appendix A) reports the structures and relevant characteristics of the analyzed compounds.

### 4.2. Instrumentation

A SCIEX API 3000-LTQ mass spectrometer (AB Sciex Framingham, MA, USA) with the standard ESI source and data system is employed for this study, and operated according to the manufacturer’s indications. The instrument employs Nitrogen from the purification system both as the nebulizer and as auxiliary gas in the ion source and as the collision gas in the q2 quadrupole.

### 4.3. Measurements

The infusion mode was used throughout this work to measure spectroscopic parameters and to analyze standards and samples. The final dilution of solutions for infusion was in 1:1 *v/v* water-methanol containing 0.1% formic acid. Flow rate of the built-in syringe was set to 10 microL × min^−1^. The concentration of standard compounds in the infused solutions varied between 0.1 and 10 microM. All experiments were carried out in the centroid mode. All compounds used in this study were characterized by means of a three-step procedure, which is described below:(a)The source spectrum of the compound or mixture was acquired, and the source conditions were optimized for maximum signal stability and intensity. This was accomplished by regulating spray gas flows and curtain plate temperature, and tuning the Declustering Potential to the value that yields the most intense signal of the desired ion species with the use of the ramp function of the instrument’s operating software.(b)For each compound, a series of fragment ion spectra was acquired in the Collision Energy ramp function mode of the instrument’s operating software, within a range of laboratory collision energies between 0 and approximately 50 to 70 V. The maximum value of collision energy depends on the molecular mass of the compound, and corresponds to a maximum of 4–6 eV_CM_, as dictated by the compound’s structure and yield of structurally informative fragments. Collision gas pressure was at the lowest setting (Low, corresponding to an ion gauge reading of 1.2 × 10^−5^ Torr (1.60 × 10^−3^ Pascal), vs. 0.8 × 10^−5^ Torr (1.07 × 10^−3^ Pascal) in the absence of collision gas).(c)The collision efficiency curves for the transitions of analytical interest were recorded both in the “forward” (Fragment Ion Scan, *see* above) and in the “backwards” (Precursor Ion Scan or Neutral Loss Scan) modes, according to the intended use of the transition, to ensure that no overlapping fragmentation processes occur.

### 4.4. Data Elaboration

Mass spectrometry measurements were extracted as spectra (ion intensity vs. *m*/*z*), XIC (ion intensity vs. time), and as Ramp curves (ion intensity vs. the relevant modified instrument voltage), and exported as .txt files from the instrument’s data system for further elaboration. Custom developed Microsoft Excel spreadsheets were used for all post-acquisition elaborations [26,27]. A few measurements were replicated across several months, in between other analytical work, to gauge robustness of the determined parameters.

### 4.5. Calculation of the Collision Energy Maximum in the Fragmentation Efficiency Curve

The value of collision energy corresponding to the maximum yield of the analytically useful fragment (or transition) was calculated, in the intensity vs. collision energy (either as laboratory, or Center-of-Mass scale) plot, as the intersection of two (least-square) straight lines, each one approximating the raising and the decreasing portion of the curve, respectively (“*teepee* method”). The interval of approximately 25% to 75% of the maximum intensity was usually selected as the linear portion for optimal least-squares calculation based on maximizing the R^2^ coefficient. The intersection point of the two straight lines was calculated from their intercept and slope values by standard analytical geometry equations. The uncertainty of the calculation was calculated from the span, on the x-axis, of the diamond-shaped intersection of the 95% confidence limit stripes of the two straight lines (Figure 4).

### 4.6. Application in the Triple Quadrupole Mass Spectrometer

The obtained value of the maximum of the collision efficiency curve for the analytically relevant fragment ion is converted to the Center-of-Mass scale (eV), and used to infer the corresponding “laboratory frame” value of collision energy for other compounds that belong to the same chemical class with the use of Equation (2). A custom Microsoft Excel spreadsheet calculates the parameters to run the *iso-*energetic Precursor Ion (*i-*PI) and Neutral Loss (*i-*NL) scans with the use of Equations (1) and (2). The layout of Scheme 5 is set to match that of the instrument’s data system, and the operator copy-and-pastes calculation results from the blue background cells directly for use. 

The actual values of the parameters reported are those of the first proof-of-principle example, see above (Section 2.2.1). The value of mTAR (cell A2) is defaulted at 28 (Nitrogen collision gas). Reference variables are typed in cells C4 (*m*/*z* of Precursor ion) and D4 (collision voltage for max. yield of fragment, interpolated from a ramp-CE experiment). The corresponding value of collision energy is calculated in E4 (formula is reported in G3). To operate the *i-*CID experiment, the desired scanning mass interval is input in cells C2 (min, START) and D2 (max, STOP), scan speed in cell E2, and collision voltage is calculated in cells F2 and G2 for CE START and CE STOP, respectively with formula in G4. Cells C2-G2 are thus directly ready for export. That reported in Scheme 2 only uses one reference compound to infer the value of CE_max_ for a series of homolog compounds. To improve accuracy, and to test whether the assumption holds for a wider series of available compounds, this spreadsheet is linked to a module (further lines below line 5 of the displayed section of the Excel spreadsheet) that calculates a more representative value of CE_max_ from results of a ramp-CE linked to the PI or CNL group scan. 

A tabular form of the complete spreadsheet with instructions is presented in Appendix B and an Excel file is available from the Author upon request. A further module of the spreadsheet (Appendix C) allows the adaption of the *i-*CID technique in case the instrument software does not allow the modulation of the (q2-Q1) collision potential synchronously with the scan of the Q1 and Q3 mass filters. In this case, a “segmented pseudo-scan” of the quadrupole filters Q1 and Q3, e.g., by 14-u “steps” can accommodate a “stepped” variation of the (q2-Q1) collision potential, with only minimal difference of collision energy from that of a continuous variation (Appendix A). Since we did not need to implement this alternative in our instrument, Appendix C reports the calculation module with the fictitious example of Figure 1 and Figure 3 (an Excel file is available from the Author upon request).

### 4.7. Derivatization of Alpha-Amino Acids as Dimethylamino-Azobenzene-Sulphonyl (Dabsyl) Amides

A mixture of alpha-amino acids, each approximately 1μM is prepared in deionized water. A 100-μL aliquot was mixed to 10 microliters of 1M sodium bicarbonate and to 100 microliters of a 10 μmol/mL (3.3 mg/mL) of dabsyl chloride in acetonitrile [23]. The mixture was vortexed for 30 s and left standing at room temperature for 2 h. An appropriate aliquot was diluted 1:100 *v/v* with the ESI infusion solvent for infusion experiments, or 10 microliters were directly injected into the HPLC-MS system for analysis. Extemporary urine from a healthy laboratory volunteer (100 μL) was treated in the same way.

### 4.8. Shotgun Separation by Liquid Chromatography of Alpha-Amino Acids Derivatized as Dabsyl Amides

A chromatographic column (Phenomenex Gemini C18, 100 × 2mm i.d., 3 mm particle size, 110 A porosity) was employed for separation. A 10 min mobile phase gradient of A (10 mM ammonium formate, 0.1% formic acid, pH 4) and B (acetonitrile) starts at 10% B for 2 min, was linearly increased to 90% B in 7 min and brought back to the initial conditions in 1 min. The column was held at 40 °C and 10 microliters were injected with the instrument’s auto-sampler. Detection by tandem mass spectrometry was achieved in the negative ion mode, by a precursor ion scan of *m*/*z* 240, scanning the m/z interval between 320 and 440 in 1 s, at the Low q2 pressure setting (approximately reading 1,2 mTorr; 1.60 × 10^−3^ Pascal). Collision energy ramp was synchronized to the Q1 scan, with a continuous variation between 22 eV_lab_ at *m*/*z* 320 and 44 eV_lab_ at *m*/*z* 440.

## 5. Conclusions

A possible advantage of the *iso-*energetic CID approach in comprehensive, or “*omic*”, bio-analyses is that more comparable fragment spectra can be obtained for closely related analytes, in which reporter fragments have closer relative intensity. Thus, quantification of target metabolites or biomarkers may improve, since the signal-to-concentration relationship will be much closer for the different terms of a series of compounds. In addition, the semi-quantitative appreciation of the level of unexpected compounds, for which calibration curves cannot be established in advance, would also be more reliable.

A recently published application of this strategy allowed characterizing some unusual ceramide glycosides that occur in pistachio and almond nuts [28]. Briefly, the sequential use of alternate *i-*PI and *i-*NL scans highlighted the presence, in the nut fatty extracts, of ceramides with a modified sphingosine linked to a hexose at their C-1 hydroxyl group. Fragment ion analysis highlighted that saturated and hydroxylated fatty acids are linked through an amide bond. This application also confirmed that the technique is compatible with scan speed used in the coupling of the triple quadrupole tandem mass spectrometer with conventional narrow bore liquid chromatography.

Several classes of natural substances, including Phase 2 xenobiotic metabolites [7] and covalent protein adducts [29,30], are similarly featured, and yield either or both common structure-specific (“*pivot*”) charged fragments (therefore prompting for the Precursor Ion approach) and neutral fragments (thus calling for the Neutral Loss approach). Given the easy availability of the additional feature of *iso-*energetic CID in the triple quadrupole tandem mass spectrometer, one prospective aim of this work is preparing a catalog of structure-specific transitions and optimized collision energies for an increasingly wide choice of organic and bio-organic classes of compounds. This information is the basis for the non-target screening and measurement of important classes of biomolecules in complex mixtures and for the discovery of new structurally related chemical entities, to incorporate into metabolomics protocols.

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
