# Peer review of "Center-of-Mass iso-Energetic Collision-Induced Decomposition in Tandem Triple Quadrupole Mass Spectrometry"

_molecules, 2020, doi:10.3390/molecules25092250_

Round 1

Reviewer 1 Report

Molecules 7761174

Center-of-Mass iso-energetic collision-induced decomposition in tandem triple quadrupole mass spectrometry.

The work is well developed. The contribution it makes is significant for an area such as the HPLCMS detection and identification of organic molecules from different sources.

Some minor suggestions are given below:

  • The author should be more specific and mention that the figure 2 in the example has been taken from the mentioned paper.

Figure 2. Re-analysis of the data by Paglia et al., 2008 (ref. 8, Figure 3)

Figure 2 (Left, taken/taken of of Paglia et al., 2008) or other paragraph

  • Section 2.2.1. Detection of dabsyl-amino acids by iso-energetic CID and Precursor Ion Scan.

The author write in Line 208,“Some amino-acid derivatives were prepared individually (list in Table S1) and”.

The series of compounds analyzed should be accompanied by the bibliography corresponding to their semisynthesis process, where their spectroscopic properties and other relevant physico-chemical properties appear for their identification.

  • Section 2.2.2. Detection of nucleosides by iso-energetic CID and Constant Neutral fragment Loss Scan.

Line 279 Protonated nucleosides (listed in Table S4)

Bibliography corresponding to their semisynthesis process should be included

  • The page numbering should be revised

After suggestions have been made, the work could be considered for publication.

Author Response

Manuscript molecules-788642 - Major Revision

Reviewer 1 MDPI | Reply review report https://susy.mdpi.com/user/manuscripts/review/11988220?report=7128149 1 Molecules 7761174

Center-of-Mass iso-energetic collision-induced decomposition in tandem triple quadrupole mass spectrometry.

The work is well developed. The contribution it makes is significant for an area such as the HPLCMS detection and identification of organic molecules from different sources.

Dear Reviewer,

First, thanks for the quick, fair and useful reading of my draft.

Some minor suggestions are given below:

The author should be more specific and mention that the figure 2 in the example has been taken from the mentioned paper.

Figure 2. Re-analysis of the data by Paglia et al., 2008 (ref. 8, Figure 3)

Figure 2 (Left, taken/taken of of Paglia et al., 2008) or other paragraph

I didn’t take Figure 2 (left) directly from Paglia’s article, since they report a (copyrighted) spectrum. I measured peak heights from a printout and used the numbers and the information in the article to draw the histogram plot and to perform the subsequent calculations (Fig.2 right). I use “the Authors” in the text to warn that actual experiments were not mine. I think that there is no ambiguity.

Is this even more explicit?

When the Authors infused an equimolar solution of the standard compounds, the resulting Precursor Ion spectrum (Figure 3 of the cited article) showed a marked and regular decrease in signal strength of the lower (C-2 to C-8) and of the higher (C-16 to C-18) homologs. Figure 2, left panel, is recalculated from the intensities in the spectrum of Figure 3 of the cited article.

Section 2.2.1. Detection of dabsyl-amino acids by iso-energetic CID and Precursor Ion Scan.

The author write in Line 208, “Some amino-acid derivatives were prepared individually (list in Table S1) and”.

The series of compounds analyzed should be accompanied by the bibliography corresponding to their semisynthesis process, where their spectroscopic properties and other relevant physico-chemical properties appear for their identification.

We prepared the few single DABS-AA from standard AA and DABS-Cl at the sub-micromole scale (0.1-0.05) according to the cited reference and directly analyzed the gross mixture by infusion-ESI, without even purification. Given the very simple chemical process involved (derivatization for a routine AA analysis in our lab), occurrence of the molecular signal was deemed sufficient to ensure compound identity and MS-MS fragments confirmed it.

Section 2.2.2. Detection of nucleosides by iso-energetic CID and Constant Neutral fragment Loss Scan.

Line 279 Protonated nucleosides (listed in Table S4) Bibliography corresponding to their semisynthesis process should be included

Ref. 22 is: Colombo F, Falvella FS, De Cecco L, Tortoreto M, Pratesi G, Ciuffreda P, Ottria R, Santaniello E, Cicatiello L, Weisz A, Dragani TA. Pharmacogenomics and analogues of the antitumour agent N6-isopentenyladenosine. Int J Cancer. 2009 May 1;124(9):2179-85. doi: 10.1002/ijc.24168.

Ref. 23 is: Ottria R, Casati S, Baldoli E, Maier JA, Ciuffreda P. N⁶-Alkyladenosines: Synthesis and evaluation of in vitro anticancer activity. Bioorg Med Chem. 2010 Dec 1;18(23):8396-402. doi: 10.1016/j.bmc.2010.09.030.

The page numbering should be revised

I cannot do it in the template (what is wrong?)

After suggestions have been made, the work could be considered for publication.

Thank you very much, best regards

FMRubino

Reviewer 2 Report

Major concerns

This manuscript proposed two scan modes of the triple quadrupole tandem mass spectrometer that can apply to determine some biological mixtures. The techniques look promising, but the comparison between conventional methods with the new method as well as the actual application are both missing, especially in the quantitative data with statistical analysis, such as ANOVA, for analyzing the biological mixtures. Hence, the demonstration of the application of this technique that the author claimed in the abstract section is not complete. In other words, we still don't know whether the methods can be applied to the analysis including characteristics as well as the quantities of biological mixtures. Their efficiency is also not clear.

Minor points

p16: face a different center-of-mass collision energy - face different center-of-mass collision energy

p25: principle examples of determination of biological mixtures - principle examples of the determination of biological mixtures

p30: collision induced dissociation - collision-induced dissociation

p50: The example refers to characterization - The example refers to the characterization

p71: sphingomyelines - sphingomyelins?

p86: some value of potential difference - some value of the potential difference

p90: of method setup - of the method set up

p256: can be identified from co-occurrence of - can be identified from the co-occurrence of

p376: Thus, detection efficiency of the different homologs - Thus, the detection efficiency of the different homologs

p395: variable nucleobase as ionic fragment - variable nucleobase as an ionic fragment

p515: structure specific - structure-specific

p519: preparing a catalogue - preparing a catalog

Author Response

Manuscript molecules-788642 - Major Revision

Reviewer 2 MDPI | Reply review report https://susy.mdpi.com/user/manuscripts/review/12000040?report=7138694 1

Dear Reviewer,

First, thanks for the quick, fair and useful reading of my draft.

Major concerns. This manuscript proposed two scan modes of the triple quadrupole tandem mass spectrometer that can apply to determine some biological mixtures. The techniques look promising, but the comparison between conventional methods with the new method as well as the actual application are both missing, especially in the quantitative data with statistical analysis, such as ANOVA, for analyzing the biological mixtures. Hence, the demonstration of the application of this technique that the author claimed in the abstract section is not complete. In other words, we still don't know whether the methods can be applied to the analysis including characteristics as well as the quantities of biological mixtures. Their efficiency is also not clear.

Reviewer 2’s major comment indeed strikes the reason why I delayed for several years now the publication of the equation and method. The two examples shown derive from “proof-of-principle”, aka aborted research plans that “customer-colleagues” prompted me to start and for which interest, materials and instrument time faded (I have at least three more, in different fields, but with an insufficient number of standard compounds available for study). The main aim of the examples in this “outing” of the general method is demonstrating that the so far never reported equation is physically right and the technique is feasible on a TSQ, in our case without compromising operation even at a comparatively fast scan rate of 0.25 s/100 Da. Our current instrument is a (now outdated) low-end triple quadrupole instrument, and the decrease in sensitivity with respect to MRM contributed to compromising the nucleotide adduct application. The current brands of our own TSQ are so more sensitive, that they can likely overcome the fall by two orders of magnitude between any scanning and MRM approach. To downplay the claim in the abstract I have rephrased as follows (no more “demonstration”):

"To exemplify the application of this technique, this article shows two proof-of-principle approaches to the determination of biological mixtures, one by Precursor Ion analysis on alpha amino acid derivatized with a popular chromophore, and the other on modified nucleosides with a Neutral Fragment Loss scan."

Of course, to develop and validate assays based on this TSQ application and on the preliminary data shown in this article will need, if any, a resumed interest from my colleagues. One on the discovery of ceramides in nuts, described in the 2020 Foods reference [26], is the only application that so far had the chance of raising a sustained interest.

Minor points

p16: face a different center-of-mass collision energy - face different center-of-mass collision energy

p25: principle examples of determination of biological mixtures - principle examples of the determination of biological mixtures

p30: collision induced dissociation - collision-induced dissociation

p50: The example refers to characterization - The example refers to the characterization

p71: sphingomyelines - sphingomyelins?

p86: some value of potential difference - some value of the potential difference

p90: of method setup - of the method set up

p256: can be identified from co-occurrence of - can be identified from the co-occurrence of

p376: Thus, detection efficiency of the different homologs - Thus, the detection efficiency of the different homologs

p395: variable nucleobase as ionic fragment - variable nucleobase as an ionic fragment

p515: structure specific - structure-specific

p519: preparing a catalogue - preparing a catalog

all corrections made in the new text.

thank you very much, best regards

FMRubino

Round 2

Reviewer 2 Report

The comments have been addressed by the author accordingly.

Author Response

Dear Reviewer 2, I attach my response as the MDPI template word file. Please see the attachment. I have accepted all phrase corrections. They are highlighted in yellow in the revised copy. As for the major comment, I agree with you and I have downplayed my statement in the abstract. Have I understood well the reason for this further revision? Best regards, FMRubino
